# Investigation of the Effect of Students’ Nodding on Their Arousal Levels in On-Demand Lectures

**DOI:** 10.3390/s23083858

**Published:** 2023-04-10

**Authors:** Kotaro Sumida, Ayumi Ohnishi, Tsutomu Terada, Hiroshi Kato, Hideaki Kuzuoka, Yoshihiko Kubota, Hideyuki Suzuki, Masahiko Tsukamoto

**Affiliations:** 1Graduate School of Engineering, Kobe University, 1-1 Rokkodaicho, Nada, Kobe 657-8501, Hyogo, Japan; kotaro-sumida@stu.kobe-u.ac.jp (K.S.); ohnishi@eedept.kobe-u.ac.jp (A.O.); tuka@kobe-u.ac.jp (M.T.); 2Faculty of Liberal Arts, The Open University of Japan, 2-12 Wakaba, Mihama, Chiba 251-8586, Chiba, Japan; hkato@ouj.ac.jp; 3Graduate School of Information Science and Technology, The University of Tokyo, 7-3-1 Hongo, Bunkyo 113-0033, Tokyo, Japan; kuzuoka@cyber.t.u-tokyo.ac.jp; 4Graduate School of Education, Tamagawa University, 6-1-1 Tamagawagakuen, Machida 194-8610, Tokyo, Japan; kubota@kubota-lab.net; 5Faculty of Humanities and Social Sciences, Ibaraki University, 2-1-1 Bunkyo, Mito 310-0056, Ibaraki, Japan; hideyuki@suzuki-lab.net

**Keywords:** on-demand lecture, sensing, heart rate information, nodding

## Abstract

Due to COVID-19, various lecture styles are being explored. On-demand lectures are attracting increasing attention due to advantages such as being able to watch without restrictions due to location and time. In contrast, on-demand lectures have disadvantages, such as no interaction with the lecturer, so the quality of on-demand lectures should be improved. Our previous study showed that when participants nod without showing their faces in a real-time remote lecture, their heart rate state changes to arousal and nodding can increase arousal. In this paper, we hypothesize that nodding during on-demand lectures increases participants’ arousal levels, and we investigate the relationship between natural and forced nodding and the level of arousal based on heart rate information. Students taking on-demand lectures rarely nod naturally, so we used *entrainment* to encourage nodding by showing a video of another participant nodding, and by forcing the participants to nod when the other participant nodded in the video. The results showed that only participants who nodded spontaneously changed the value of pNN50, an index of the arousal level, which reflected a state of high arousal after one minute. Thus, participants’ nodding in on-demand lectures can increase their arousal levels; however, the nodding must be spontaneous, not forced.

## 1. Introduction

Due to the COVID-19 pandemic, lecture styles other than face-to-face lectures are becoming more important. In particular, recently, universities around the world have been distributing lectures through OpenCourseWare [1], and the value of on-demand lectures is expected to increase.

On-demand lectures have various advantages, such as being able to watch without restrictions due to location and time, and participating in lectures from various universities and well-known lecturers. On the other hand, unlike face-to-face lectures and real-time remote lectures, on-demand lectures have disadvantages, such as no interaction with the lecturer and difficulty in maintaining motivation [2]. Hence, the quality of on-demand lectures has to be improved. To improve the quality of on-demand lectures, it is important to improve the lecture itself, such as optimizing the lecture content for on-demand use. In addition, if devising an on-demand lecture environment for students improves the quality of on-demand lectures, then such an approach should be applied.

In our previous study of real-time remote lectures [3], when students’ faces were visible on the application in a real-time remote lecture, students’ nodding could be used as an assessment behavior, which had the same meaning as nodding in a face-to-face lecture. On the other hand, in the experiment, when students turned off the camera and nodded without showing their faces, and because the other person was not watching their face, it was not seen as an assessment behavior targeted at others. In addition, their change in heart rate indicated that they had become aroused. In other words, nodding in response to a lecture without being seen by other students can increase the students’ level of arousal, resulting in the quality improvement of lectures. The effect of such nodding without others can change other mental states besides the level of arousal. Therefore, we hypothesized that nodding in on-demand lectures, in which no one watches the students in the lecture, would have a similar positive effect on their arousal level and other mental states as it does when the students’ faces are invisible in real-time remote lectures. However, our previous study [3] has shown that nodding without being seen by others is unusual for many students, so even if nodding has an arousal effect, such nodding needs to be encouraged to take advantage of this effect.

For this reason, in this study, we used a method to encourage natural nodding by having the students of an on-demand lecture watch a nodding video while watching the lecture. An image of this method is shown in Figure 1.

In this method, to encourage the student’s nodding, the student watches a lecture video containing a video of another student nodding to the same lecture as the student. This phenomenon of being caught in motion by the motion of others is called *entrainment* [4], and entrainment is known to occur with nodding during a remote conversation [5]. In addition, we investigated how arousal levels and other mental states change when students are forced to nod as well as when students nod naturally. This is because if forced nodding has an arousing effect, the quality of on-demand lectures can be enhanced by using a system that forces nodding in on-demand lectures. If forced nodding is ineffective, and only natural nodding is effective, then a mechanism to induce natural nodding is needed.

Based on the above, in this paper, we collect nodding and heart rate information from participants in on-demand lectures to examine how nodding affects their arousal levels and whether nodding also affects their other mental states. When investigating arousal, we focused on the effect of nodding on the participants themselves and divided the participants into three conditions: viewing the lecture as usual, viewing the lecture as usual with a video of another participant nodding to the lecture, and viewing the lecture with a video of another participant nodding to the lecture and forcing them to nod, as well as viewing another participant’s nods in the video. This study clarifies whether nodding in on-demand lectures is better and whether the nodding should be done naturally or forced by some form of external approach. Based on the results, we discuss ways to improve the quality of remote lectures.

The contributions of this study are the following:We conducted an experiment simulating an actual on-demand lecture with a total of 31 participants, and analyzed heart rate information when participants nodded in the on-demand lecture under three conditions: (1) attending the lecture as usual, (2) attending the lecture as usual with a video of another participant nodding to the lecture that causes *entrainment*, and (3) attending the lecture with a video of another participant nodding to the lecture and forcing them to nod when the other participant nods in the video.From the results of the experiment, we confirmed that the students’ own arousal levels increased when they nodded spontaneously in the on-demand lecture. In addition, we showed the possibility of improving the quality of on-demand lectures by encouraging students to nod spontaneously.

The remainder of this paper is structured as follows: Section 2 introduces related research, Section 3 describes the validation method, and Section 4 presents the experimental results. The results are discussed in Section 5, and the paper is concluded in Section 6.

## 2. Related Research

### 2.1. Research on Remote Lectures

Various types of remote lectures are available, including on-demand, real-time, and real-time lectures with or without the students’ faces shown. Many studies have investigated improvements in the quality of real-time remote lectures. Spathis et al. used the attention tracking function of Zoom [6], a remote meeting system, to evaluate the performance of real-time remote lectures by scoring the students’ attention [7]. They also investigated how to use the attention-tracking and recording functions available in Zoom in a face-to-face setting. The results showed that no correlation was found between attention and grades, and additional hardware and software are needed to use Zoom’s functions in face-to-face lectures, indicating that Zoom may still be utilized in terms of functionality even when face-to-face lectures can be conducted. The attention-tracking feature of Zoom is currently disabled due to privacy concerns. Prasanth et al. used Microsoft Teams [8] to analyze the facial expressions and head movements of the audience during a real-time remote presentation and devised a system called *AffectiveSpotlight*, in which the most responsive and expressive person is visible to the presenter, making it easier for the presenter to speak online [9]. Cacault et al. studied the effects of real-time remote lectures on student attendance and academic performance, and found that overall attendance decreased slightly and academic performance increased for the ablest students and decreased for the least able students [10].

Although various studies have investigated how to improve the quality of remote lectures, no studies have investigated the mental states of participants, such as the arousal level, using objective indicators such as sensors to improve the quality of remote lectures by improving the mental states of participants. Therefore, this study aims to improve the quality of remote lectures by focusing on the relationship between the learner’s nodding and heart rate information.

### 2.2. Research on Effects of Nodding

This study investigates the effects of nodding during lectures on biological information. Nodding has various effects, and many studies have investigated the effects of nodding on others. Matarazzo et al. investigated the effects of the nodding of an interviewer during an interview on the interviewee [11]. They reported that interviewees who were interviewed by an interviewer who nodded increased their speaking time more than those who were interviewed by an interviewer who did not nod. Osugi et al. found that the nodding of a Computer Graphics (CG) character that imitated a human woman increased the familiarity and likability of the CG character for the viewers of the CG character [12]. Maeda et al. attempted to improve the speaker’s ease of speaking by remotely tracking the audience’s gaze and nodding in real time and projecting it on the screen to visualize the audience’s reactions and interests [13]. Fujii et al. found that in an on-demand VR lecture using Head-Mounted Displays, students felt an atmosphere of involvement and positivity among them after viewing avatars of students performing nodding motions in the VR space [14]. Thus, nodding to others is known to have various positive effects.

Nodding is also known to cause entrainment, and studies on the entrainment of nodding have also been reported. Sakata et al. examined how the entrainment of nodding by the listener differs between face-to-face and on-screen situations by dividing the conversation into two groups: the speaker and the listener [5]. The results showed that entrainment occurs both face-to-face and through a screen, although the brain regions that are active during entrainment are different.

Thus, previous studies have mainly investigated the effects of nodding on others, and have not investigated the effects of nodding on the actor’s biological information. In our previous study, we investigated the effects of nodding during lectures on the actors’ biological information, and the results suggested that nodding during remote lectures is not only for others but also for the actors themselves, especially when the participants’ faces are invisible on the lecture tool, and that such nodding can increase arousal [3]. Therefore, based on the hypothesis that nodding affects the actor’s biological information obtained in our previous study, to investigate how nodding affects their arousal level and other mental states, this study aims to collect nodding and heart rate information from participants in an on-demand lecture.

## 3. Method of Verification

### 3.1. Experimental Conditions

This study aims to examine the effects of nodding on the arousal level and other mental states of students in on-demand lectures and to investigate whether students should nod even in on-demand lectures for higher-quality lectures, and how such nodding should be achieved. For this purpose, we collected the participants’ nodding and heart rate information. To collect data on nodding, this experiment encouraged nodding by having the participants watch a lecture with a nodding video. In this method, to encourage the participants’ nodding, participants watched a lecture video containing a video of another participant nodding to the same lecture as the participants, which causes *entrainment*. We hypothesized that the effects of nodding would differ between the nodding encouraged by this method, the nodding forced on the participant, and the nodding in a normal lecture without this method, so we designed an experimental condition to compare these three types of nodding.

The experimental conditions were divided into the following three: attending the lecture as usual (*no nodding video*), attending the lecture as usual with a video of another participant nodding to the lecture (*with nodding video/not forced*), and attending the lecture with a video of another participant nodding to the lecture and forcing them to nod, as well as another participant nodding in the video (*with nodding video/forced*). Particularly in the condition of *with nodding video/forced*, we instructed the participants to nod at the same time as the other participant before the lecture. No specific instructions were given in the other conditions. In this experiment, one of the authors performed the role of the other participant who nodded in the video.

The lecture videos for each condition are shown in Figure 2. Figure 2a is a video of a typical on-demand lecture. Participants in the *no nodding video* condition watched this video. Figure 2b is a typical on-demand lecture video, with another participant, as shown in the upper-right corner of the figure, watching the lecture in advance and nodding in response to the lecture. Participants in the *with nodding video* condition watched this video. In this video, another participant nodded when he understood or agreed with what the lecturer said. Another participant’s position did not overlap with the text of the lecture content, so he did not interfere with the lecture as much as possible. The lecturer’s face was always visible in the video.

### 3.2. Verification Methods

The participants’ environment is shown in Figure 3. In the experiment, the participants’ faces were captured by a webcam attached to a personal computer, and they viewed the lecture while wearing a WHS-3 wearable heart rate sensor from the Union Tool Corporation [15] on their chests.

To compare the experimental conditions, participants were divided into three groups, and all participants attended the same lecture (first half 30 min, rest 20 min, and second half 60 min). The three groups were as follows:

Group I: First half: *no nodding video*, Second half: *no nodding video*

Group II: First half: *no nodding video*, Second half: *with nodding video/not forced*

Group III: First half: *no nodding video*, Second half: *with nodding video/forced*

In all three groups, the first half of the condition was the same, to confirm individual differences in the heart rate information and responses to the lecture.

The participants were undergraduate and graduate students, with 9 in Group I and 11 each in Group II and Group III, for a total of 31 (30 males and one female). Participants watched the lecture in the same environment in which they usually attended lectures at any location for both the first and second halves. The on-demand lecture was conducted by a professor at the Open University of Japan, who was an expert in giving on-demand lectures. The content of the lecture was related to sociology, and the difficulty and the amount of information in the first and second halves of the lecture were made to be as equal as possible. A questionnaire survey was conducted after the experiment. This experiment was conducted with the approval of the Ethical Review Committee for Research Directly Involving Human Subjects at the Graduate School of Engineering, Kobe University.

### 3.3. Evaluation Methods

To confirm the effects of participant nodding on arousal and other mental states, we collected heart rate information and nodding data from the participants in this experiment. After collecting data on nodding, we observed the timing of the participants’ nodding along with pNN50, which is suitable for focusing on changes in a short time among heart rate information, and investigated the effect of nodding on the actor’s own heart rate information. In addition, a questionnaire survey was conducted to assess the subjective evaluation of the participant’s arousal level and so on.

#### 3.3.1. Heart Rate Information

In this study, we analyzed the autonomic nervous system to understand participants’ mental states, such as arousal and concentration, and what happens to participants when they nod. The heart rate information allows us to analyze the activity level of the autonomic nervous system function. The autonomic nervous system is divided into two types: sympathetic and parasympathetic. Generally, concentration, arousal, and mental load are high when the sympathetic nervous system is active and low when the parasympathetic nervous system is active [16,17,18]. As for heart rate information, we used two indicators, LF/HF (Low Frequency/High Frequency) values and pNN50 (percentage of the difference between adjacent normal-to-normal intervals greater than 50 ms) values, which were calculated from heart rate intervals (RRI). LF/HF is the sympathetic nerve activity that can be calculated from the RRI [19,20]. When the sympathetic nervous system is active, the LF component appears, whereas the HF component decreases, resulting in a large value of LF/HF. In the experiment, LF/HF was calculated from the past two minutes of heart rate data, as it shows high reliability in long-term analysis. Therefore, we estimated the trend of the students’ arousal level throughout the entire lecture from the LF/HF. pNN50 is the ratio of consecutive adjacent heartbeats whose RRI exceeds 50 ms, reflecting the activity of the parasympathetic nervous system. As with LF/HF, pNN50 is often used as an index of the arousal level [21,22,23]. When the parasympathetic nervous system is active, the value of pNN50, which is highly responsive and suitable for focusing on changes in parasympathetic nerve activity in a short time, increases. In this study, we used pNN50 to analyze the heart rate information in combination with the timing of nodding that occurred in a short time. We used the WHS-3 series, which is a wearable heart rate sensor, to acquire heart rate information such as LF/HF and pNN50. The sensor can acquire heart rate information, such as RRI, LF, and HF. From the RRI, we could calculate the pNN50. The pNN50 was calculated for the past 100 samples, which corresponded to a period of approximately one minute in the past. The sampling frequency of the electrocardiogram was 1 kHz. The heart rate sensor calculated the RRI, LF, HF, and other values based on the collected electrocardiogram information and sent them to the mobile device.

#### 3.3.2. Nodding

Because nodding was detected with low accuracy in automatic detection by video analysis, we counted nods through manual labeling by observing the facial images of each participant using an annotation tool, ELAN [24]. As a criterion for counting nods, nods that were responses to the lecture were counted, rather than any movement of the head. Nods are sometimes performed with multiple movements of the head in a single movement, but these movements were counted as a single nod.

#### 3.3.3. Nodding Effect

To confirm whether nodding increased the arousal level, we analyzed the heart rate information when participants nodded. Specifically, we compared the time variation of pNN50 when participants nodded. An image of the analysis method is shown in Figure 4. The red dots in the figure represent the time at which the participant nods. The pNN50 at the red point when nodding occurred uses the RRI of the interval that does not include nodding as the calculation interval. In contrast, the pNN50 at the black point uses the RRI of approximately one minute after the moment of nodding as the calculation interval, so it includes the heart rate information at the moment of nodding in the calculation interval. Therefore, by comparing the two points, the point of pNN50 at the time of nodding and the point of pNN50 one minute later, we compare the changes in the heart rate information of participants before and after nodding. This change in heart rate information between the two points before and after nodding is hereafter defined as the *nodding effect*. If the value of the *nodding effect* is negative, pNN50 is tilted in a negative direction and the participant is considered to show increased arousal by nodding.

In the calculation of the *nodding effect*, we consider the original pNN50 tendency of the participant and use a method that can calculate the effect of nodding without participants’ original pNN50 influence as much as possible. If, for example, the pNN50 is increased due to sleepiness, the effect of nodding will reflect the original sleepiness of the participant, but if the effect of the original slope of the pNN50 can be eliminated as much as possible, the effect of the nodding itself can be observed. Based on the hypothesis that nodding without others is done for oneself and such nodding can increase arousal, which is suggested by the authors’ previous study, the slope $N_{s}$ of pNN50 between the pNN50 at the moment that the participant nodded (defined as *t*) and the pNN50 approximately one minute later after the moment that the participant nodded (defined as $t_{0}$) is obtained as follows.
$$N_{s}=\frac{\mathrm{pNN}50\left(t+t_{0}\right)-\mathrm{pNN}50\left(t\right)}{t}$$

From this value, we subtract the slope $N_{all}$ of the approximate straight line of the entire graph of the student’s pNN50 to obtain the *nodding effect N*, which is defined as follows.
$$N=N_{s}-N_{all}$$

This calculation method was devised based on the fact that pNN50 is calculated for approximately the past one minute, and assumes that the effect of nodding appears in the following minute, from the nodding moment. Moreover, by subtracting the slope of the approximate line from its value, the effect of the original tendency of pNN50 can be ignored.

#### 3.3.4. Questionnaire Survey

To confirm the subjective mental state of the participants, a questionnaire survey was conducted for each condition to determine the subjective arousal level and concentration level of the participants. A questionnaire survey was conducted after the end of the entire lecture. The first half of the questionnaire consisted of the following items, and each answer was rated on a 5-point scale.

Was the lecture difficult? (5: difficult, 3: adequate, 1: easy)Did you understand the lecture? (5: understood, 3: neither, 1: not understood)Were you able to concentrate on the lecture? (5: concentrated, 3: neither, 1: not concentrated)Were you sleepy in the lecture? (5: sleepy, 3: neither, 1: not sleepy)

In the second half of the survey, in addition to the same content as the first half of the questionnaire, the following two items were surveyed for the participants under the condition of *with nodding video*. Moreover, they were asked to write free comments on the lecture.

Were you distracted by the video of the author? (5: distracted, 3: neither, 1: not distracted)Were you caught by the video of the author? (5: caught, 3: neither, 1: not caught)

## 4. Results and Discussion

### 4.1. Investigation of the Effect of Nodding on the Arousal Level of the Participants

#### 4.1.1. Investigation of the Change in the Number of Nods in Each Group

We counted the number of nods from each participant during the lecture, and examined changes in the number of nods between lecture conditions. Nods were collected by labeling the participants’ face videos with ELAN. In Group I, two out of nine participants, and in Group III, one out of 11 participants failed to record facial videos, so we analyzed data from a total of 28 participants, including 11 participants in Group II. The timing of nodding in the first half of the lecture (approximately 30 min) and the second half (approximately 60 min) is shown in Figure 5 and Figure 6. In Figure 5, the results are arranged from top to bottom as Group I, Group II, and Group III. In Figure 6, the timing of another participant’s nodding in the video is given at the top, and the results for each group are arranged below the timing of another participant’s nodding. The horizontal dots indicate the timing of the expression in each lecture condition. The red dots indicate when participants nodded and the light blue line indicates when participants were asleep. The dots in each partition are lined up vertically according to the number of participants. In Group I, the nine participants were designated A–I; in Group II, the 11 participants were designated J–T; and in Group III, the 11 participants were designated U–AE. Because we failed to record facial videos of participants H and I in Group I and participant AE in Group III, we discuss the other participants. In Group I, participants did not nod at all in both the first and second halves, so we assumed that no difference existed between the first and second halves due to the change in the lecture content. In Group II, some participants began to nod or nodded more in the second half, so the nodding video is considered to have encouraged nodding through entrainment. However, this entrainment was not observed in all participants in Group II. In Group III, all participants nodded in the second half because they were instructed to nod in sync with the video nodding. In this group, no nod was observed except when another participant nodded, meaning that no spontaneous nodding was observed by the participants. We cannot determine whether this situation can be called entrainment. However, to verify whether such forced nodding increases arousal, we will discuss this situation with heartbeat information.

#### 4.1.2. Evaluation of the Nodding Effect

To verify whether nodding increased arousal, we analyzed heart rate information when participants nodded. Specifically, we compared the temporal changes in pNN50 when participants nodded. The graphs of the participants in Group II and Group III who nodded most frequently are shown in Figure 7. The red dots in the figure indicate when participants nodded. The blue dotted line indicates the approximate straight line of pNN50. When the value of pNN50 is large, parasympathetic activity is activated, and when it is small, parasympathetic activity is suppressed. The pNN50 is calculated for the past 100 samples, which corresponds to approximately one minute of past time.

Based on the hypothesis that nodding without others is done for oneself and such nodding can increase arousal, which is suggested by our previous study, the *nodding effect* was calculated using the calculation method described in Section 3.3.

The *nodding effect* was calculated from 70 nods for three participants who nodded in the not forced condition (*no nodding video, with nodding video/not forced*), two data from the first half and two data from the second half. The calculated mean value was −3.9 ×$10^{-4}$ (/s), so we confirmed that the mean value of the *nodding effect* was negative. The *nodding effect* was calculated for 1339 nods for 10 participants who nodded when forced, from 10 data from the second half. The calculated mean value was 6.6 ×$10^{-4}$ (/s), so we confirmed that the mean value of the *nodding effect* was positive. Thus, we confirmed that the mean value of the *nodding effect* without being forced is negative, while the mean value when forced is positive.

Finally, we statistically validated this result by comparing the interval in which the *nodding effect* occurred and the interval in which no nodding occurred. Specifically, we calculated this with the same method as the *nodding effect*, using the instantaneous values of all the intervals in which each participant did not nod, and estimated the trend of pNN50 in the interval in which participants did not nod. We performed a Kruskal–Wallis test using each value as a sample under three conditions: the value obtained from the above calculation (the trend of pNN50 without the nodding interval), the value of the *nodding effect* under the not forced condition, and the value of the *nodding effect* under the forced condition. The results of the test are shown in Figure 8. The test results showed a significant difference (H=235.15,df=2,p<0.1). Therefore, we performed multiple comparisons using the LSD method and found that the trend of pNN50 without the nodding interval was significantly larger than the *nodding effect* value under the not forced condition (T=2.367,p<0.05). Thus, if nodding is spontaneous, participants can attend a lecture at a higher arousal level than they would normally. In other words, we believe that a device enabling students to nod naturally in on-demand lectures will increase their arousal levels, and, therefore, improve the quality of on-demand lectures. The value of the *nodding effect* under the forced condition was significantly larger than the pNN50 trend without the nodding interval and the *nodding effect* under the not forced condition (T=15.034,p<0.05), (T=6.529,p<0.05). Thus, forced nodding causes lower arousal than when attending a normal lecture or attending a lecture with spontaneous nodding. These results indicate that the forced nodding itself has a high load, and that nodding against understanding makes it more difficult to understand and makes the participant sleepy.

The above results suggest that although nodding without others can increase arousal, the nodding must be done spontaneously, not forced. This consideration also suggests that nodding should be encouraged in on-demand lectures. However, because nodding via some form of external approach is expected to have negative effects on students, encouraging students to nod naturally will improve the quality of lectures the most. In addition, as regards lecture methods, instructing students to listen to a lecture without falling asleep is difficult, but instructing students to listen to a lecture while nodding is easy from both the instructor’s point of view and the students’ point of view. Thus, for this reason, the effect of nodding can be useful.

### 4.2. Correlation or Causation?

In this study, we assumed that nodding increases arousal. However, on the other hand, we could also interpret that nodding increases due to high arousal levels. Since the data obtained in this experiment do not indicate whether nodding comes first or arousal comes first, we have no firm indication as to whether a simple correlation exists between nodding and arousal, or whether a causation exists in which nodding promotes arousal, as hypothesized in our study. However, in our previous study [3], which was the basis for the hypothesis that nodding encourages arousal, in the face-to-face lecture, the heart rate did not change significantly to a state of arousal by nodding, which is an evaluative behavior. This means that normal nodding toward others did not encourage arousal, nor did normal nodding occur when the arousal level was high. Moreover, in previous studies examining the effects of nodding, no results were found indicating that people nodded when their arousal level was high. Therefore, based on previous studies, we do not consider the hypothesis that people nod when their arousal level is high, which is the opposite of the interpretation of this study.

For this reason, the experimental results of the nodding effect are described below based on the hypothesis that nodding increases arousal.

### 4.3. Investigation of the Effect of Nodding on the Mental State of the Participants Other Than Nodding Effect

#### 4.3.1. Investigation of the Participants’ Arousal Tendencies throughout the Entire Lecture

To confirm how the lecture styles affected the students’ arousal levels, we analyzed the tendencies of the participants’ arousal levels throughout the entire lecture based on heart rate information. Specifically, for each lecture style, we calculated the average value of LF/HF for the participants throughout the entire lecture. A mixed two-factor analysis of variance was performed using LF/HF as the sample for each group and the first and second half under the 3 × 2 condition. However, no significant results were found. Therefore, in this verification, we cannot conclude that the lecture style changed the students’ level of arousal.

#### 4.3.2. Discussion of the Statements Obtained from the Questionnaire Survey

Although no significant results were found in the numerical analysis of the questionnaire survey, we discuss the results of this experiment based on the comments obtained from the free descriptions.

Participants in Group II, who were placed under the *with nodding video/not forced* condition, reported positive comments such as “It felt more like a face-to-face lecture”, “The fact that other participants’ videos were shown was different from a normal on-demand lecture, and it felt like a real-time lecture”, “The fact that other participants’ faces were shown made me want to listen properly/wake up”, and “I felt uncomfortable with the video only at the beginning, but it did not mind it at all in the latter half of the lecture.”

The participants in Group III, who were placed under the *with nodding video/forced* condition, reported positive comments such as “I felt it was easier to understand, probably because it was easier to understand the main points”, “It felt more like a real-time lecture than a normal on-demand lecture”, “I felt more like listening to the lecture because I was consciously trying to nod”, and “ I felt a sense of unity and it improved my drowsiness.” On the other hand, negative comments such as “It was difficult to listen to the lecture while looking at the author’s face” and “It was a little difficult to understand the content of the lecture because I was conscious of matching the timing of nodding” were also obtained from this group.

These comments suggest that the method of showing a nodding video can have a positive effect on the students as long as nodding is not forced.

## 5. Discussion

Based on the results of Section 4, this section discusses the current limitations of this experiment, the problems found in the experiment, and the improvements based on these discussions.

### 5.1. Limitation

Regarding the method of estimating the effect of nodding, this paper used pNN50, a type of heartbeat information commonly used as an index of the arousal level. Although the present results focused on the relationship between pNN50 and the arousal level, a different interpretation is possible because pNN50 is known to be an indicator of the arousal level as well as the tension level and mental load. Therefore, to further investigate the possibility that nodding without others can increase arousal, we need to confirm the results from multiple experiments, including other indicators.

Regarding the effect of nodding on the actors themselves, this study assessed the effect of nodding using original calculations based on the hypothesis obtained in our previous study. In this study, considering the window size of pNN50, we assumed that the effect of nodding appears approximately one minute after nodding. We consider this calculation interval of one minute to be somewhat appropriate because it corresponds to the interval for which pNN50 was calculated in this study. However, neither how long the effect of nodding on the actor lasts nor whether the change in heart rate information occurs before or after nodding is unknown. Since no previous research on the relationship between nodding and heart rate information was found, for example, by varying the window size of pNN50 and making comparisons on several patterns of window size, further investigation and analysis are needed to determine whether the present calculation method is correct or whether another analysis method would be better.

Regarding the method used to encourage participants to nod, two methods were used in this experiment: we encouraged nodding during the lecture by showing a video of another participant’s nodding using nodding entrainment, and we forced participants to nod when another participant in the video nodded. In this paper, the former is defined as not forced, and the latter as forced. However, even with the former method, some participants possibly felt that they were forced to nod because they were shown a video of another participant nodding. In addition, regarding the latter method, the participants possibly nodded spontaneously when another participant nodded. Such nodding needs to be distinguished from non-forced nodding, but there was difficulty in interpreting which of these nods the participant was nodding to from the video, and we have not been able to distinguish them. Therefore, in the next experiment, we need to distinguish nodding by asking the participants whether they were nodding spontaneously in the questionnaire or by having the participants label the nodding themselves.

Regarding the approach used in our on-demand lectures, we believe that it could be adapted to other online lecture formats. For example, nodding when a student’s face is not shown in a remote real-time lecture has potential to increase the level of arousal and improve the quality of the lecture experience.

To confirm whether nodding promotes arousal, it is necessary to analyze the changes in arousal from the timing of nodding. As we have done in previous studies and in this study, the simplest way to confirm the phenomena is to count the timing of nodding and look at the change in arousal level at that time. As an index of the arousal level, visual information such as flicker measurement and detailed subjective evaluation data can be considered, but, in this study, we focused on heart rate information, which is commonly used.

### 5.2. Improvements and Future Experiments

In future experiments, to verify the effect of nodding in on-demand lectures, we need to collect other biological data such as body movements and eye contact, and expressions other than nodding to reconsider the indicators and analysis methods, and to be able to distinguish between natural and forced nodding, as points to be improved based on what was described in Section 5.1.

Moreover, to encourage participants to nod more naturally in future experiments, we will use Avatarify [25]. Avatarify is a tool that can add arbitrary actions to facial videos captured by a camera. By modifying Avatarify to add behaviors such as nodding in real time to facial videos hereafter, Avatarify enables us to collect more natural nodding using video self-modeling [26]. Video self-modeling is a method of encouraging action by watching oneself performing the action. By showing a video of participants themselves nodding in real time, we can collect more data on nodding. Therefore, we will use Avatarify in future experiments to verify the results of the present study based on more nodding.

## 6. Conclusions

This paper defines a unique value for the *nodding effect*, and investigates the relationship between nodding and pNN50 to improve the quality of on-demand lectures. This investigation examined whether students should nod in on-demand lectures, and whether such nodding should be spontaneous or forced by some form of external approach. Based on the results of the experiment, we discuss how to improve the quality of remote lectures.

The results of the investigation confirmed the possibility that when participants nodded voluntarily, their nodding increased their arousal level. This result suggests that nodding without others can have the effect of increasing arousal, but the nodding must be spontaneous and not forced. Through this consideration, we can conclude that nodding should be encouraged in on-demand lectures; however, because nodding via some form of external approach is expected to have negative effects on students, encouraging students to nod naturally will improve the quality of lectures the most.

In the future, we plan to conduct further experiments to encourage more nodding naturally by showing a video of the students themselves nodding, rather than the author, and continue to collect a sufficient amount of data to clarify the relationship between nodding and heart rate information.

## Figures and Tables

**Figure 1 sensors-23-03858-f001:**
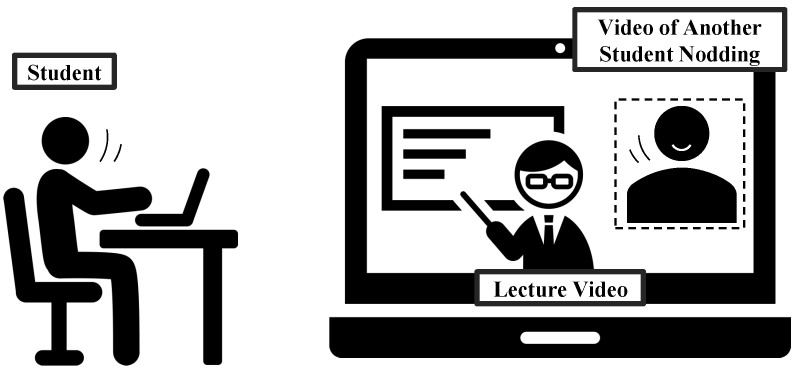
Image of the method to encourage natural nodding.

**Figure 2 sensors-23-03858-f002:**
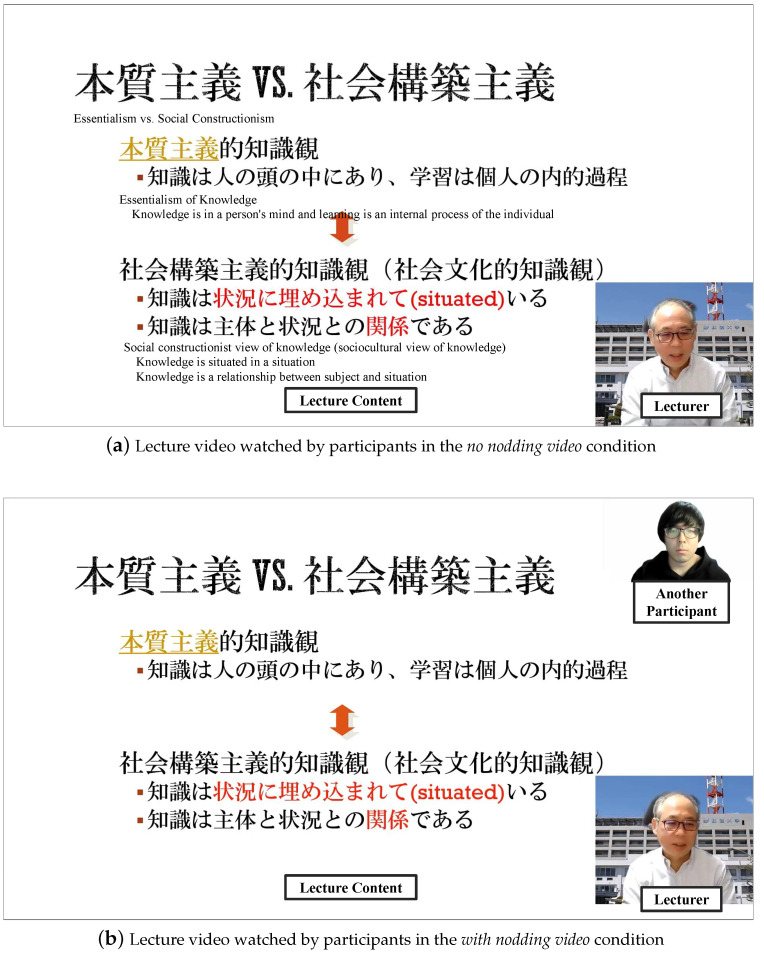
Lecture videos for each condition.

**Figure 3 sensors-23-03858-f003:**
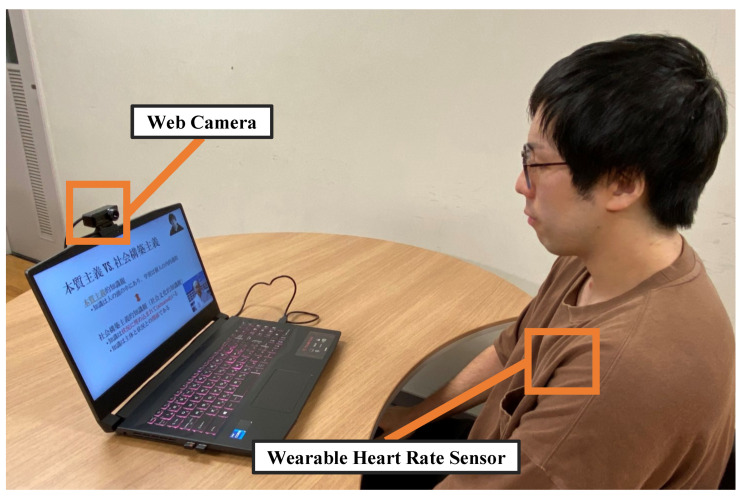
The participants’ environment.

**Figure 4 sensors-23-03858-f004:**
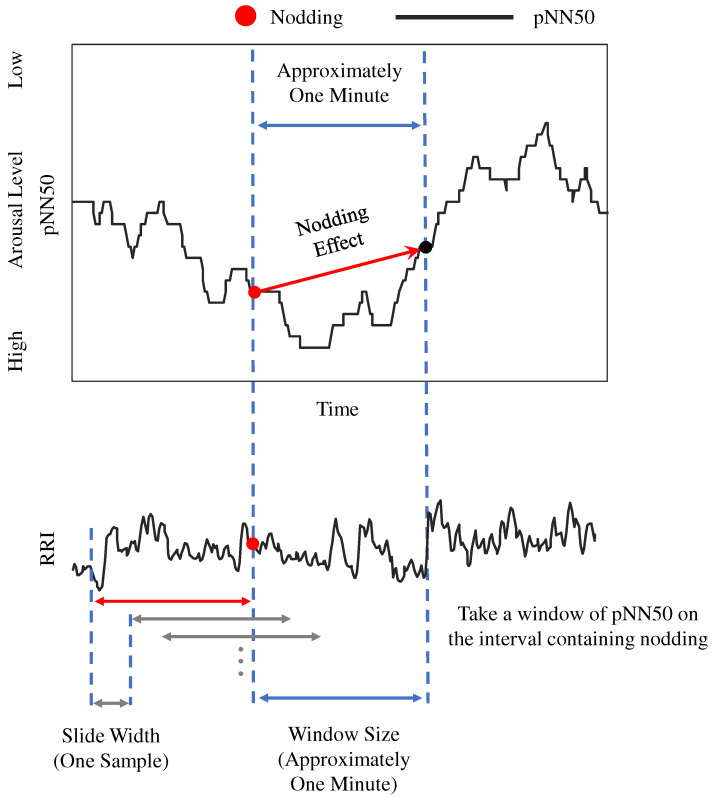
Nodding effect.

**Figure 5 sensors-23-03858-f005:**
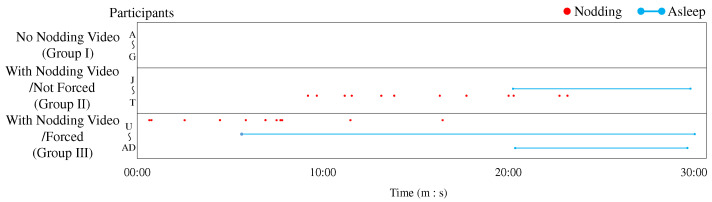
Time series data of nodding in the first half of the lecture.

**Figure 6 sensors-23-03858-f006:**
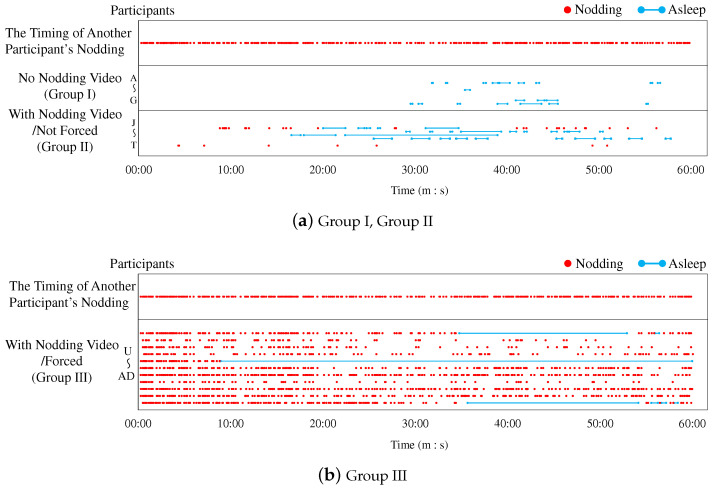
Time series data of nodding in the second half of the lecture.

**Figure 7 sensors-23-03858-f007:**
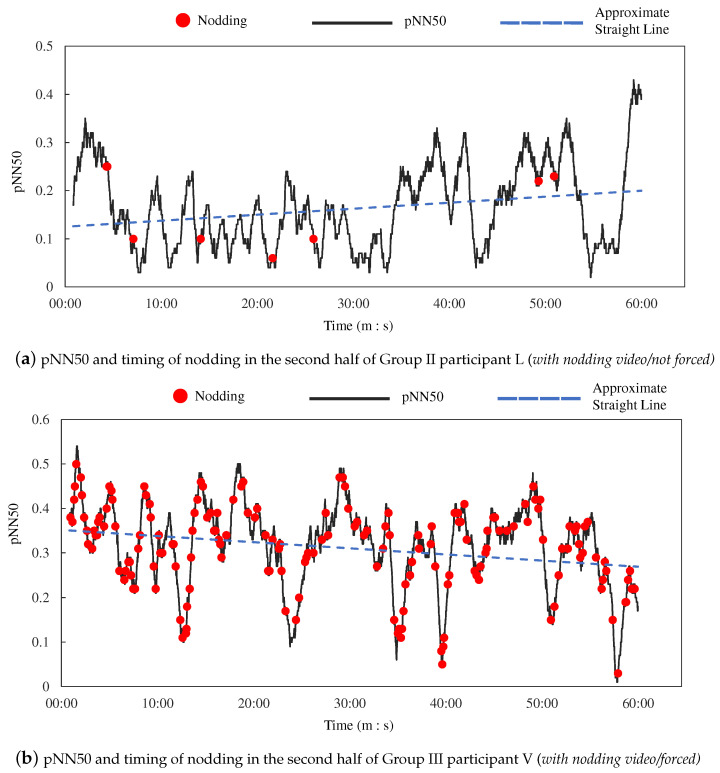
pNN50 and timing of nodding by participants who nodded a lot.

**Figure 8 sensors-23-03858-f008:**
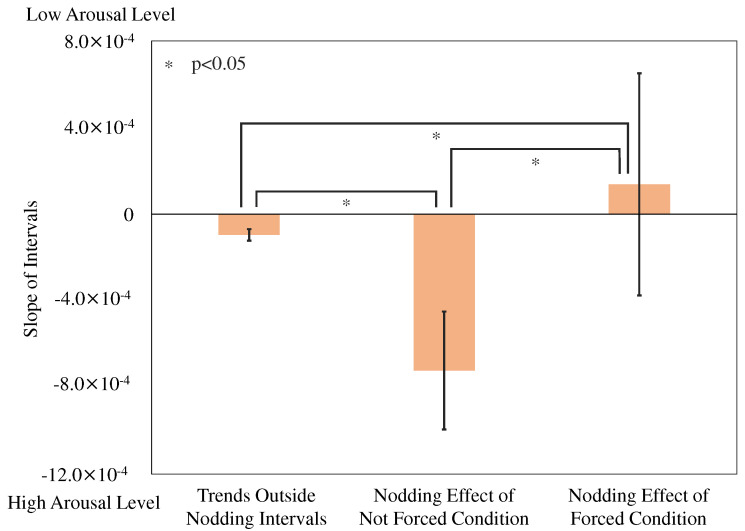
Results of the test of the slope of the interval.

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
