# Peer review of "Investigation of the Effect of Students’ Nodding on Their Arousal Levels in On-Demand Lectures"

_sensors, 2023, doi:10.3390/s23083858_

Round 1
Reviewer 1 Report
Dear Authors,
Happy to review you article.
Please address following points to improve the article.
Line 1-2: “easy viewing” may be replaced with appropriate words. Also in other places.
Line 33-35: Difficult to understand.
Line 115: CG not explained.
Line 120: HMDs not explained.
Line 173: PC not defined
Please pay attention to all the abbreviations and explain first time they are used.
The methods of verification section should be organized better to include all the statistical test being used against all the hypothesis.
Accordingly organize the sections of results, discussion and conclusion.
Regards,
Author Response
Response to Reviewer 1 Comments
First of all, we would like to thank again the reviewers for their useful comments, suggestions, and criticisms. Below, we provide response to each comment and the improvements done in the revised version of the paper.
Point 1: Line 1-2: “easy viewing” may be replaced with appropriate words. Also in other places.
Response 1:
We apologize for any inappropriate wording. We originally intended to convey the meaning of "being able to watch lectures" with the phrase "easy viewing," but we have revised the wording to more accurately convey this intention.
Point 2: Line 33-35: Difficult to understand.
Response 2:
We apologize for the unclear phrasing. We have reworded the relevant section and provided a suitable format to describe the experimental conditions.
Point 3: Line 115: CG not explained.
Point 4: Line 120: HMDs not explained.
Point 5: Line 173: PC not defined.
Response 3-5:
We added explanations of the abbreviation to each word.
Point 6: The methods of verification section should be organized better to include all the statistical test being used against all the hypothesis.
Response 6:
We thank you for your suggestion. The hypothesis in this paper is that nodding during on-demand lectures could increase the arousal level of participants. To examine this hypothesis, we analyzed participants' heart rate information during nodding, and our findings indicate that participants' spontaneous nodding indeed increases their arousal level. All statistical tests conducted are included in the above verification.
Reviewer 2 Report
The article is very interesting and suitable for any reader in this field. However, some recommendations should be considered for publication:
11. Evaluate the possibility of including the section 2. Related Research in the section 1. Introduction.
22. 3. Method of verification: consider include an outline about the experimental method carried out. Include the different experimental conditions.
33. Discussion: Clarify this section, one option is creating a quantitative comparative study of the results obtained.
44. Discussion: From point of view of this reviewer, the limitation of pNN50 and improvements and future experiment are not part of this section.
55. Include a quantitative comparative to demonstrate the limitation of pNN50.
66. Would be possible compare the sensibility of the pNN50 sensor with other sensor type?
77. Improve the conclusion include the “highlights” of results and discussion.
Minor recommendations:
1. The abstract is too long. A single paragraph of about 200 words maximum.
2. Check references, there are mistakes. They do not adapt to the MDPI format. For example, the correct format of paper is: Author 1, A.B.; Author 2, C.D. Title of the article. Abbreviated Journal Name Year, Volume, page range.
3. Include more references about this field, for example:
Küchler, A.-M.; Schultchen, D.; Dretzler, T.; Moshagen, M.; Ebert, D.D.; Baumeister, H. A Three-Armed Randomized Controlled Trial to Evaluate the Effectiveness, Acceptance, and Negative Effects of StudiCare Mindfulness, an Internet- and Mobile-Based Intervention for College Students with No and “On Demand” Guidance. Int. J. Environ. Res. Public Health 2023, 20, 3208. https://doi.org/10.3390/ijerph20043208
Vergara-Rodríguez, D.; Antón-Sancho, Á.; Fernández-Arias, P. Variables Influencing Professors’ Adaptation to Digital Learning Environments during the COVID-19 Pandemic. Int. J. Environ. Res. Public Health 2022, 19, 3732. https://doi.org/10.3390/ijerph19063732
Author Response
Response to Reviewer 2 Comments (Round 1)
First of all, we would like to thank again the reviewers for their useful comments, suggestions, and criticisms. Below, we provide response to each comment and the improvements done in the revised version of the paper.
Point 1:
Evaluate the possibility of including the section 2. Related Research in the section 1. Introduction.
Response 1:
We thank you for your suggestion. We considered organizing it together in response to your comment. However, we finally divided Section 1 and Section 2 into two separate sections in this paper because we thought that including Section 2, which is approximately one page in length, within Section 1 would make it too long.
Point 2:
- Method of verification: consider include an outline about the experimental method carried out. Include the different experimental conditions.
Response 2:
As you point out, we agree that writing a summary is helpful to understand the experimental overview too. Although we considered writing a summary, the required information is already provided in sections 3.1 and 3.2 of this paper. Including a summary would result in duplicated content, which would make it difficult to read.
Based on your comment, we realized that the titles "Experimental Conditions" and "Experimental Methods" in Sections 3.1 and 3.2, respectively, could cause confusion. Therefore, we revised the title of Section 3.2 to "Verification Methods" to provide better clarity and make it easier to grasp the outline of those two sections.
Point 3:
Discussion: Clarify this section, one option is creating a quantitative comparative study of the results obtained.
Response 3:
We appreciate your suggestion. As mentioned in Section 2.2, previous studies examining the effects of nodding have stated that nodding improves the familiarity of CG characters, that communication is activated by the display of nodding, and that speaking time increases when the interviewer nods.
However, this paper proposes a novel idea that nodding can also promote arousal, which makes it difficult to compare our findings with previous studies.
Point 4:
Discussion: From point of view of this reviewer, the limitation of pNN50 and improvements and future experiment are not part of this section.
Response 4:
Because discussion section is written from the viewpoint of discussing what have been found from the experiments, the authors think that future experiments and improvements should be described here.
However, at the beginning of the section, we put a more detailed description, such as "This section discusses the current limitations of this experiment, the problems found in the experiment, and the improvements based on these discussions.”
Point 5: Include a quantitative comparative to demonstrate the limitation of pNN50.
Point 6:
Would be possible compare the sensibility of the pNN50 sensor with other sensor type?
Response 5 and 6:
The pNN50 and LF/HF are commonly used values as indices of arousal. We have added references and explanations in the second paragraph of section 3.3 so that this is clear from the paper.
References:
Narumon, J.; Tipporn, L.; Midori, S. A Preliminary Evaluation of Comfortable Arousal using Biological Information Measurement for Autonomous Driving, In Proceedings of Asia Pacific Conference on Robot IoT System Development and Platform, Online, 29--30 November 2021; pp. 83--84.
Evgeny, G.V; Marsha, E.B.; Bronya, V.; Paul, L.; Tomoko, U.; Eun, Y.M.; Suchismita, R. Heart Rate Variability Response to Alcohol, Placebo, and Emotional Picture Cue Challenges: Effects of 0.1-Hz Stimulation, Psychophysiology 2008, 45, 847--858.
Suzuki, K.; Iguchi, T.; Nakagawa, Y.; Midori, S. A Prototype of Multi-modal Interaction Robot Based on Emotion Estimation Method Using Physiological Signals, In Proceedings of Asia Pacific Conference on Robot IoT System Development and Platform, Tokyo, Japan, 1--2 November 2022; pp. 7--12.
Point 7:
Improve the conclusion include the “highlights” of results and discussion.
Response 7:
Thank you for pointing this out. We added the summary of results to the Conclusions section, as it was not clearly described in the initial version of the paper.
Regarding the summary of the discussion, we are focused on discussing future perspectives. The main contribution of this paper is the proposal and validation of the novel idea that nodding promotes arousal. We believe that the most important point of this discussion is the need to implement and analyze practices based on the ideas presented in this paper in various fields. We have briefly discussed this point in the last paragraph in Conclusions.
Point 8:
The abstract is too long. A single paragraph of about 200 words maximum.
Response 8:
We counted the number of words again and the abstract is exactly 198 words. So we kept it as it was.
Point 9:
Check references, there are mistakes. They do not adapt to the MDPI format. For example, the correct format of paper is: Author 1, A.B.; Author 2, C.D. Title of the article. Abbreviated Journal Name Year, Volume, page range.
Response 9:
We apologize for the mistakes. We revised the formatting of the bibliography.
Point 10:
Include more references about this field, for example:
Küchler, A.-M.; Schultchen, D.; Dretzler, T.; Moshagen, M.; Ebert, D.D.; Baumeister, H. A Three-Armed Randomized Controlled Trial to Evaluate the Effectiveness, Acceptance, and Negative Effects of StudiCare Mindfulness, an Internet- and Mobile-Based Intervention for College Students with No and “On Demand” Guidance. Int. J. Environ. Res. Public Health 2023, 20, 3208. https://doi.org/10.3390/ijerph20043208
Vergara-Rodríguez, D.; Antón-Sancho, Á.; Fernández-Arias, P. Variables Influencing Professors’ Adaptation to Digital Learning Environments during the COVID-19 Pandemic. Int. J. Environ. Res. Public Health 2022, 19, 3732. https://doi.org/10.3390/ijerph19063732
Response 10:
Thank you for the introduction. I have thoroughly reviewed the articles you mentioned. However, we have decided not to include them in the papers due to their being based on questionnaires rather than sensor-based data collection, which is the focus of our research.
However, we responded to this comment by increasing the overall number of papers cited, for example by comments from other review comments.
Reviewer 3 Report
It’s a pleasure to review this article with the practical contribution. Some suggestions are proposed as follows.
1. Core objective of this research is to improve the quality of on-demand lectures. To this aim, the authors mentioned devising the lecture and the environment. It is suggested to give a broader picture before concentrating on the proposed approach. Similarly, before emphasizing the nodding effects, the authors could introduce possible helpful biological information.
2. The authors could comment on the characteristics of the experimental on-demand lecture to apply research findings to similar courses.
3. For the technical detail of the experiments, the authors might consider presenting the stepwise procedures and related measurements in a flowchart or table.
4. Since the authors conducted questionnaire surveys during the experiment, they could present the results more in-depth and comment on their implications. In addition, learning performance might need some verification.
5. Discussing the correlation or causation between nodding and arousal is insightful. Other than the results of the authors’ previous studies, it would be helpful to give some theoretical support.
6. The authors could support the analyses of improvements and future experiments with a more evident rationale. Besides, they better describe how to decide the sufficient data to clarify the relationship between nodding and heart rate information.
Author Response
Response to Reviewer 3 Comments (Round 1)
First of all, we would like to thank again the reviewers for their useful comments, suggestions, and criticisms. Below, we provide response to each comment and the improvements done in the revised version of the paper.
Point 1:
Core objective of this research is to improve the quality of on-demand lectures. To this aim, the authors mentioned devising the lecture and the environment. It is suggested to give a broader picture before concentrating on the proposed approach. Similarly, before emphasizing the nodding effects, the authors could introduce possible helpful biological information.
Response 1:
This paper is based on a hypothesis derived from our previous research, and we recognize that the general approaches to improving on-demand classes have not been presented clearly in a top-down style and have appeared more bottom-up in structure. However, we believe that presenting the details of the hypothesis in sections 1 and 2 is crucial to understanding the experiments in this paper.
As a response to this comment, we added a discussion in Section 5 that refers to common arousal indicators and explains the possibility of their verification. We also explained why we focused on the nodding effect, making the discussion more understandable. Specifically, we added the following points:
To confirm whether nodding promotes arousal, it is necessary to analyze the changes in arousal from the timing of nodding. As we have done in previous studies and in this study, the simplest way to confirm the phenomena is to count the timing of nodding and look at the change in arousal level at that time.
As an index of arousal level, visual information such as flicker measurement and detailed subjective evaluation data can be considered, but in this study, we focused on heartrate information, which are commonly used.
Point 2:
- The authors could comment on the characteristics of the experimental on-demand lecture to apply research findings to similar courses.
Response 2:
As you suggested, the approach used in our on-demand lectures could be adapted to other online lecture formats.
For example, nodding when a student's face is not shown in a remote real-time lecture has potentially increase the level of arousal and improve the quality of the lecture experience. As the viewpoint of this, we have included the discussion in Section 5.
Point 3:
For the technical detail of the experiments, the authors might consider presenting the stepwise procedures and related measurements in a flowchart or table.
Response 3:
In this experiment, we collected heart rate information and recorded instances of nodding to investigate whether nodding promotes arousal.
Although these can be summarized in a table, in the present article we wanted to focus most on the nodding effect, and the importance levels of the indicators are too different from each other. We did not add a table because we thought that the summarized table would mislead people into thinking that all indicators were of equal importance in parallel.
Point 4:
Since the authors conducted questionnaire surveys during the experiment, they could present the results more in-depth and comment on their implications. In addition, learning performance might need some verification.
Response 4:
The results to be discussed from the questionnaire are described in Section 4.3.2. Although the number of paragraphs could be increased, we have summarized them in one section in order to focus on the part of the data analysis where new and clearer findings were obtained.
With regard to the learning performance, as Goertler et al. (2018) stated that learning outcomes such as test results did not differ from those in the case of face-to-face lectures in online language education using Zoom in combination with remote real-time and on-demand lectures, the experiment has already been conducted. Therefore, it was not the subject of this experiment.
Point 5:
Discussing the correlation or causation between nodding and arousal is insightful. Other than the results of the authors’ previous studies, it would be helpful to give some theoretical support.
Response 5:
As explained in Section 2.2, previous studies investigating the effects of nodding have stated that it enhances the familiarity of computer-generated(CG) characters, activates communication when nodding is displayed, and increases speech time during interviews. However, these studies did not extensively analyze the biological data and did not confirm whether people actually nod when their arousal level is high. Although heart rate has been identified as an arousal index, there is no prior research that analyzed nodding in combination with it. Hence, we believe that our study indicates interesting results.
Point 6:
The authors could support the analyses of improvements and future experiments with a more evident rationale. Besides, they better describe how to decide the sufficient data to clarify the relationship between nodding and heart rate information.
Response 6:
To be sufficient data requires consideration of quantity and quality factors. We can state what is necessary, but it is difficult to say for sure what is sufficient until we have done the experiments and confirmed the results.
The following is a summary of what we know as far in advance as possible.
Here is a summary of what we currently know: For the experiment we conducted with 30 participants, the data obtained should be sufficient to some extent for a single experiment. However, due to the variability in lecture styles, it is necessary to gather data on several typical patterns, such as different subjects and schools.
As for the qualitative component of the data to be taken, as mentioned in the first paragraph of the constraints, we believe that the results will be more reliable if data other than pNN50 are taken as supportive indicators of arousal (e.g., visual information such as flicker measurement, detailed subjective evaluation data, etc.). We added to the constraints the details of the data to be taken as described above.
Round 2
Reviewer 2 Report
the authors have increased the quality of the manuscript. From this reviewer's point of view, this manuscript can be published in its current version. Congratulations!